# Disrupting *Bordetella* Immunosuppression Reveals a Role for Eosinophils in Coordinating the Adaptive Immune Response in the Respiratory Tract

**DOI:** 10.3390/microorganisms8111808

**Published:** 2020-11-17

**Authors:** Monica C. Gestal, Uriel Blas-Machado, Hannah M. Johnson, Lily N. Rubin, Kalyan K. Dewan, Claire Bryant, Michael Tiemeyer, Eric T. Harvill

**Affiliations:** 1Department of Infectious Diseases, College of Veterinary Medicine, University of Georgia, Athens, GA 30602, USA; Hannahmariejohnson@gmail.com (H.M.J.); lilrubin@gmail.com (L.N.R.); kkd112014@gmail.com (K.K.D.); harvill@uga.edu (E.T.H.); 2Department of Pathology, Athens Veterinary Diagnostic Laboratory, University of Georgia, Athens, GA 30602, USA; ublas@uga.edu; 3Department of Veterinary Medicine, University of Cambridge, Cambridge CB30ES, UK; ceb27@cam.ac.uk; 4Complex Carbohydrate Research Center, Department of Biochemistry and Molecular Biology, University of Georgia, Athens, GA 30602, USA; mtiemeyer@ccrc.uga.edu

**Keywords:** eosinophils, *Bordetella bronchiseptica*, adaptive immunity, immunomodulation

## Abstract

Recent findings revealed pivotal roles for eosinophils in protection against parasitic and viral infections, as well as modulation of adaptive immune responses in the gastric mucosa. However, the known effects of eosinophils within the respiratory tract remain predominantly pathological, associated with allergy and asthma. Simulating natural respiratory infections in mice, we examined how efficient and well-adapted pathogens can block eosinophil functions that contribute to the immune response. *Bordetella bronchiseptica*, a natural pathogen of the mouse, uses the sigma factor *btrS* to regulate expression of mechanisms that interfere with eosinophil recruitment and function. When *btrS* is disrupted, immunomodulators are dysregulated, and eosinophils are recruited to the lungs, suggesting they may contribute to much more efficient generation of adaptive immunity induced by this mutant. Eosinophil-deficient mice failed to produce pro-inflammatory cytokines, to recruit lymphocytes, to organize lymphoid aggregates that resemble Bronchus Associated Lymphoid Tissue (BALT), to generate an effective antibody response, and to clear bacterial infection from the respiratory tract. Importantly, the failure of eosinophil-deficient mice to produce these lymphoid aggregates indicates that eosinophils can mediate the generation of an effective lymphoid response in the lungs. These data demonstrate that efficient respiratory pathogens can block eosinophil recruitment, to inhibit the generation of robust adaptive immune responses. They also suggest that some post-infection sequelae involving eosinophils, such as allergy and asthma, might be a consequence of bacterial mechanisms that manipulate their accumulation and/or function within the respiratory tract.

## 1. Introduction

Eosinophils are well-known to be involved in asthmatic [1,2,3,4,5,6,7,8,9,10] and allergic reactions [11,12,13,14,15]. However, the eosinophilic dysregulation that contributes to these disorders does not likely reflect the authentic evolutionarily conserved roles of these unique cells in maintaining host health [14,16,17,18,19,20]. Eosinophils are recognized to play crucial roles in immune responses to parasites [16,21,22,23,24,25,26], as well as in the maintenance of homeostasis [27,28,29,30] and regulation of inflammatory responses in the gastrointestinal mucosal epithelium [24,25,26,27,28,29,30,31,32]. In fact, eosinophils have antibacterial [33,34] capabilities themselves, including the ability to engulf bacteria [35,36,37,38,39,40,41,42,43,44], or to form traps [45,46,47] that physically contain bacteria, similar to other immune cells, such as neutrophils. However, how eosinophils modulate adaptive immune responses during bacterial respiratory infections remains not fully understood [48]. Intriguingly, there is a correlation between respiratory infections and increasing risk of asthma and allergies [49,50,51,52,53,54], suggesting that eosinophils might be involved in the immune responses to these infections, and infection-induced dysregulation may contribute to this hyper-responsiveness. However, if the role(s) of eosinophils in the immune response to infection involves complex interactions and communication with multiple other immune cells, it/they might be difficult to detect and/or study in crude models of human pathogens infecting mice. In addition, the more important these cells might be to a vigorous immune response, the stronger the selective pressure on well-adapted pathogens to subvert their functions, obscuring our ability to measure and study them.

The *Bordetella bronchiseptica*–mouse experimental system is particularly useful as a natural infection model that allows for the detailed study of the interactions between bacteria and host immune system, each of which can be manipulated both independently and in combination, to probe and define their molecular-level interactions. Using this powerful experimental system, we recently discovered a *B. bronchiseptica* sigma factor (*btrS*) which is involved in the modulation of host immune responses, inhibiting T- and B-cell-mediated immunity to allow bacteria to persist indefinitely. In its absence, the host immune response is much more robust and can completely clear the infection [55]. Importantly, the resulting robust immune response both clears these otherwise lifelong infections and prevents subsequent infections with *B. bronchiseptica* or the closely related species *B. pertussis* [55]. The induced protection is stronger than that induced by vaccination or prior infection with the wild-type parental strain; this is further evidence of a coordinated immunosuppressive strategy that is *btrS*-dependent. This experimental system now allows for an exploration of the improved protective immunity induced by the mutant, to better understand the immunological mechanisms involved that are able to better control and clear infection, which has important applications to ongoing efforts to improve the currently failing pertussis vaccines.

Transcriptomic data of macrophages challenged with *B. bronchiseptica (Bb),* or a D*btrS* mutant derivative (BbD*btrS*), revealed that cytokines associated with eosinophil activation and asthma, such as IL-25, IL-33, and CXCL10, were suppressed via a *btrS*-dependent mechanism [55]. We further observed that *btrS* mediates the suppression of eosinophil influx to the lungs during *B. bronchiseptica* infections. In the absence of *btrS,* eosinophils are recruited to the respiratory tract in large numbers, and lymphoid cells aggregate in the lungs. Mice lacking eosinophils were severely defective in the recruitment of T and B cells and their organization into recognizable lymphoid aggregates, and they were deficient in the production of IL-17, IL-6, and IL-1b, which are cytokines that are known to be critical for the clearance of *Bordetella* spp. [56,57,58,59,60,61,62,63,64]. Together, these findings reveal that *B. bronchiseptica* blocks eosinophil influx via a *btrS*-dependent mechanism(s), to prevent eosinophil-mediated recruitment and organization of lymphoid aggregates that resemble BALT and stronger protective immunity. Altogether, these results suggest that prior difficulties in detecting eosinophil contributions to the generation of adaptive immunity might be because successful respiratory pathogens have evolved means to suppress eosinophil functions.

## 2. Materials and Methods

### 2.1. Bacterial Strains and Culture Conditions

Bordetella bronchiseptica RB50 wild-type (Bb) and RB50 mutant strain, herein BbDbtrS [55], were grown on plates of Difco Bordet-Gengou agar (BD, cat. 248200) supplemented with 10% sheep defibrinated blood and 20 μg/mL concentration of streptomycin, as previously described [55,65].

### 2.2. Animal Experiments

Wild-type C57BL/6J, Balb/cJ (WT) and Balb/cJ DdblGATA-1 [66] (EO^-^) mice were obtained from Jackson Laboratories, Bar Harbor, ME, or our breeding colony (established from Jackson laboratories mice). Mice were bred and maintained at Paul D. Coverdell Center for Biomedical and Health Sciences, University of Georgia, GA, (AUP: A2016 02-010-Y2-A3) [55]. All experiments were carried out in accordance with all institutional guidelines (Bordetella Host Interactions AUP: A2016 02-010-Y2-A6). Nasal cavity, trachea, lungs, spleen, and blood were collected, post-mortem, in 1 mL of cold Phosphate-buffered saline (PBS). When tissues were used to enumerate colonies, collection was performed in beaded tubes. When organs were collected for immunological studies, they were collected in 15 mL falcon tubes containing sterile PBS.

For pathological studies, mice were euthanized by using CO_2_, followed by fixation with Formalin [55,67]. Tissues were subsequently processed and stained with hematoxylin and eosin. The undersigned board-certified pathologist performed, blindly, all microscopic evaluations of these hematoxylin and eosin (HE)-stained sections at the Athens Veterinary Diagnostic Laboratory. Immunohistochemistry was performed as follows. Briefly, the primary antibody used was non-conjugated polyclonal rabbit anti-CD3 antibody (Dako A0452). The antibody was provided as a ready-to-use antiserum with staining time of 60 min. The antigen retrieval performed was heat-induced epitope retrieval, using citrate buffer with a pH of 6.0 (HK086-9K, Biogenex, San Ramon, CA, USA), for 15 min. Endogenous peroxidase was blocked by using 3% hydrogen peroxide (H312-500, Fisher Scientific, Fair Lawn, NJ, USA), for 5 min. All other blocking was completed with Power Block (HK085-5K, Biogenex, San Ramon, CA, USA), for 5 min. Positive tissue controls consisted of formalin-fixed, paraffin-embedded mouse lymph node. As a negative control, the primary antibody was eliminated and substituted with Universal Negative Control provided as the immunoglobulin fraction of serum from non-immunized rabbits. Streptavidin conjugated to horseradish peroxidase antibody conjugate system (4+HRP, HP604 H, L; Bio Care Medical) in PBS (K1016, Dako, Carpinteria, CA, USA) for 10 min was used as a detection for the secondary antibody. The substrate–chromogen system used was 3, 3′ diaminobenzidine (DAB; K3466, Dako, Carpinteria, CA, USA), for 12 min. The tissue sections were counterstained with Gills #2 hematoxylin, followed by bluing, and then they were dehydrated in ethyl alcohol levels of 70%, 95%, and 100%, cleared in xylene, and mounted with xylene-based mounting medium. 

### 2.3. Enzyme-Linked Immunosorbent Assays and Cytokine Assays

Serum samples were utilized to evaluate antibody titers. Then, 96-well microtiter plates (Costar) were coated with heat-killed *Bb*Δ*btrS* as previously reported [55]. SureBlue (SeraCare, cat. 5120-0076) was added to start the reaction, which was terminated with HCl, after three minutes. The plates were read at an OD of 450 nm. The titer was determined to be the reciprocal of the lowest dilution in which an OD of 0.1 was obtained.

To asses IL-1β, TNF-α, IL-17, and IL-6 expression, the collected trachea was homogenized, using the bead mill. Following homogenization, the tubes were centrifuged for 5 min, at 14,000 rpm and 4 °C, and the supernatant was collected to quantify levels of cytokines via commercially available ELISA kits (DuoSet ELISA system, R&D Systems).

### 2.4. Flow Cytometry

Spleen and lungs were processed and stained as previously described [55,68,69,70]. Numbers of live cells were enumerated with Countess II (Thermo Fisher) with trypan blue stain. Two million live cells were seeded in each well, for staining (see Table 1, below). The acquisition of the data was performed by using BD-LSR II (Becton Dickinson), and the analysis was completed with FlowJo 10.0, following standard gating strategy. Statistical significance was calculated by using two-way ANOVA in GraphPrism.

We used fluorescence minus one (FMO), unstained control, and compensation beads controls in each one of our experiments. Dilutions were titrated prior to the assay, to guarantee no unspecific stains.

Our gating strategy included first selecting the single cells population, using Forward scatter height versus forward scatter area density plot for doublet exclusion (FSCH/FSC-A); then, with the selected population, we performed a doublet discrimination (SSCW/SSC-H); and, finally, we cleaned the sidescatter by using SSC-A/SSC-H. We then followed the manufacture’s recommendation to specifically select viable cells, using Ghost dye (Tonbo, 13-0871-T500). From there, we continued our grating strategy by following the criteria shown in Table 2.

### 2.5. Ethics Statement

This study was carried out in strict accordance with the recommendations in the Guide for the Care and Use of Laboratory Animals of the National Institutes of Health. The protocol was approved by the Institutional Animal Care and Use Committee at the University of Georgia, Athens (A2016 02-010-Y2-A3 Bordetella-Host Interactions and A2016 07-006-Y2-A5 Breeding protocol). All animals were anesthetized with 5% isoflurane and euthanized by using carbon dioxide inhalation, followed by cervical dislocation, to minimize animal suffering. Animals were handled by following institutional guidelines, in keeping with full accreditation from the Association for Assessment and Accreditation of Laboratory Animal Care International.

### 2.6. Statistical Analysis

All results were graphed in GraphPrism (version 8.0.2), and statistical significance was calculated by using two-way ANOVA. A power analysis was used (G-Power 3) to compare the immunological response and colonization, and we utilized at least 6 mice per group (and experiments were performed in triplicate), at alpha 0.05 and power of 80%.

For all of our experiments, we used both genders (male and female), to account for gender variability.

## 3. Results

### 3.1. Splenic eosinophils Numbers Increase in Response to B. bronchiseptica

Our prior transcriptomic analysis of macrophages exposed to wild-type Bb or BbD*btrS* strains revealed that Bb suppresses the production of IL-33, IL-25, CXCL10, and other factors associated with inflammation [55]. Based on these results, we hypothesized that *btrS* mediates the modulation of inflammatory cell recruitment and activation in the respiratory tract [55]. To investigate this possibility, we inoculated C57BL/6J mice with the wild-type strain of Bb, and, at different time points, we enumerated bacterial and immune cell numbers in the lungs and spleen. Bb numbers rose rapidly, as expected, and this infection increased eosinophil (CD11b^+^CD193^+^SiglecF^+^ cells) numbers in the spleen, slightly, by day three, and over three-fold by day seven (Figure 1A), coinciding with the peak of bacterial numbers (Figure 1B). However, eosinophil numbers were only slightly increased in the lungs (Figure 1C), despite the large numbers of bacteria there (Figure 1B). The large increase in spleen but not at the site of infection suggested respiratory infection results in a substantial increase in eosinophil numbers produced in the spleen, but Bb somehow interferes with their recruitment to the lungs.

### 3.2. Bordetella bronchiseptica Blocks Eosinophil Recruitment to the Lungs

Since the *Bb*D*btrS* mutant failed to suppress the macrophage inflammatory response, based on our previously published transcriptomic data, and induced much more effective protective immunity in vivo [55], we tested the hypothesis that *Bb* blocks eosinophil recruitment to the lungs by a *btrS*-mediated mechanism. Consistent with this hypothesis, mice challenged with *Bb*D*btrS* had substantially more eosinophil numbers in both the spleen (Figure 1A) and lungs (Figure 1C), coinciding with the peak of infection (Figure 1B). Moreover, the numbers of eosinophils at the site of infection remained high until *Bb*D*btrS* was cleared from the lungs (Figure 1B,C), suggesting that *Bb* uses a BtrS-dependent mechanism to block eosinophils’ recruitment (Figure 1C). Furthermore, the timing of their recruitment, which peaked just before *Bb*D*btrS* numbers began to decline, implicated these eosinophils as potential mediators of the robust immunity induced by the *Bb*D*btrS* strain.

### 3.3. Eosinophil-Deficient Mice Fail to Clear Respiratory Infection

To investigate the contribution of eosinophils to the robust immune response to *Bb*D*btrS*, we compared infections in wild-type (Balb/cJ) and eosinophil-deficient (ΔdblGATA-1) mice, which fail to produce mature eosinophils and are one of the most commonly used models for the study of eosinophilic functions and biology [66]. Wild-type (WT) mice were able to rapidly clear bacteria from the upper and lower respiratory tract (Figure 2). In contrast, eosinophil-deficient mice (EO^-^) more slowly cleared bacteria from the lower respiratory tract (Figure 2C) and failed to clear bacteria from the upper respiratory tract, even after 169 days (Figure 2A,B). Interestingly, when WT and EO^-^ mice were challenged with wild-type *Bb*, no differences in the colonization dynamics were detected (Appendix A), revealing that *Bb* blocks both eosinophil recruitment and function via *btrS*-mediated mechanism. When EO^-^ mice were challenged with *Bb*D*btrS*, the colonization timescale resembled that caused by wild-type *Bb* (Appendix A), indicating that, in the absence of eosinophils, *btrS* plays a small role in infection. Overall, these results suggest that *Bb* blocks eosinophil-mediated generation of protective immunity via a *btrS*-dependent mechanism.

### 3.4. Eosinophils Contribute to Cytokine Secretion in Respiratory Mucosal Surfaces

The failure of the eosinophil-deficient mice to clear *Bb*D*btrS* infection suggests that eosinophils contribute to the generation of protective adaptive immunity within the respiratory tract, analogous to their role in gastric mucosal immunity, also demonstrated in these mice [71]. To explore how eosinophils augment the robust adaptive immunity to *Bb*D*btrS*, we inoculated this strain into WT and EO^-^ mice and assessed cytokine levels in the trachea, which is rich in mucosal surfaces [28], and, where such differences can be measured [26], IL-1b and IL-17, key cytokines for the clearance of *Bordetella* spp. infections [57,64,72,73,74], were increased in WT mice at day seven post-challenge (Figure 3A,B). IL-6 was increased in WT mice at day 14 post-infection, while, in EO^-^ mice, these differences were not detectable (Figure 3C). These results suggest that the milieu of signals instigated by the highly immunogenic *Bb*D*btrS* strain are substantially missing in the absence of eosinophils, implicating eosinophils in the cytokine response that shapes the generation of adaptive immunity.

### 3.5. Eosinophils Mediate Lymphocyte Recruitment to the Lungs

The markedly different cytokine profiles produced by these two different mouse strains suggest that B- and T-cell recruitment might be modulated by eosinophils. It is known that eosinophils can present viral antigen to CD8 T cells [48], and there is evidence that they also modulate B cells [75,76]. Moreover, the modulatory effect of eosinophils in B and T cells can be observed even in naïve uninfected mice [48,75,77]. We have previously reported that the T- and B-cell numbers in the lungs of WT mice were at least twice as high on day 14 post-challenge with *Bb*D*btrS* as in WT mice challenged with *Bb* [55]. To examine the effect of eosinophils on the substantial T- and B-cell recruitment induced by *Bb*D*btrS*, we inoculated this strain into WT or EO^-^ mice. T-cell numbers were substantially higher in WT Balb/c mice by day 14 post-infection, as in C57BL/6J mice (Figure 1), while in EO^-^ mice, there were smaller numbers that increased modestly with infection but were still substantially lower than uninfected WT (Figure 4A). B-cell numbers were recruited to the lungs by *Bb*D*btrS* in even higher numbers in WT Balb/c mice, but their numbers were unchanged by this severe infection in EO^-^ mice (Figure 4B). While in the wild-type mice the numbers of B cells nearly tripled by 14 days post-challenge, in eosinophil-deficient mice, B-cell numbers remained much lower and did not change measurably in response to 14 days of infection. This substantial defect in the recruitment of B cells to the lungs of EO^-^ mice suggests that eosinophils are critical to the robust local B-cell response to this infection.

### 3.6. Eosinophils Are Required for an Efficient Antibody Response

To investigate whether the defect of B cells’ numbers in the lungs of EO^-^ mice is reflective of systemic differences in B-cell response, we measured anti-*Bb*D*btrS* antibody titers in the serum of wild-type and EO^-^ mice, at various days after inoculation with *Bb*D*btrS*. In WT mice, *Bb*D*btrS*-specific antibody titers increased to over 1000 by day 21 and over 4000 by day 28. In stark contrast, EO^-^ mice barely generated detectable antibody titers by day 21, and values had only risen to about 100 by day 28, even though the bacterial numbers, reflecting antigen load, were sustained at much higher numbers than in WT mice (Figure 5A,B). These data indicate that eosinophils play a crucial role in systemic antibody production in response to respiratory infection, which can explain the delay in clearance of the bacterial infection from the respiratory tract.

### 3.7. Eosinophils Contribute to an Organized Local Immune Response

We anticipated that failure to generate a robust adaptive response would lead to differences in the lung inflammatory responses. In order to investigate how the local inflammatory and cellular response might be affected by eosinophils, we challenged WT and EO^-^ mice to examine changes in the induced lung pathology. Wild-type mice developed organized inflammatory cell-mediated responses (Figure 6A), with formation of lymphoid aggregates assembled into recognizable BALT. EO^-^ mice had less lymphoid activity, despite having higher bacterial number, and formed only indistinct structure, characterized by fewer, smaller, and less organized accumulations of macrophages and neutrophils, with few lymphocytes (Figure 6B), suggesting defective generation of localized immune response in these eosinophil-deficient mice. These distinct responses indicate that eosinophils can regulate both the inflammatory response and the generation of protective immunity by mediating the formation of lymphoid aggregates resembling BALT, in response to local respiratory infections*,* analogous to their role in the gastric mucosa [28].

## 4. Discussion

We recently demonstrated that *btrS*-mediated mechanisms contribute to *B. bronchiseptica*’s ability to suppress immunity to persist in the respiratory tract [55]. Here we demonstrate that *btrS*-mediated mechanisms suppress the accumulation and function of eosinophils, identifying these cells as targets of immunosuppression. More detailed examination of eosinophil-deficient mice revealed profound defects in aspects of the stimulation and organization of functional protective adaptive immunity. EO^-^ mice were defective in the ability to produce key cytokines, recruit lymphocytes, and organize them in aggregates in which local adaptive immunity can be generated. In addition to revealing important new roles for eosinophils, this work suggests that targeted disruption of eosinophil functions can suppress the local immune response in the respiratory tract. Other persistent commensals and pathogens are likely to have evolved similar strategies, potentially explaining why these important roles of eosinophils were not previously readily observed; successful respiratory pathogens can block them.

The particulars of the role(s) of eosinophils in adaptive immunity in the respiratory tract are intriguing. One hypothesis is that, during bacterial respiratory infections, eosinophils facilitate immune homeostasis and adaptive immunity [78] by influencing the cytokine milieu and immune signaling pathways [79,80,81,82,83,84]. This would generate a strong selective pressure on bacteria to evolve and develop mechanisms to block eosinophilic functions. Excitingly, we report a new role for eosinophils which involves the generation of robust protective immunity against bacterial respiratory infections, apparently mediated by the formation of lymphoid aggregates near the site of infection. A better understanding of the molecular mechanisms by which eosinophils mediate the recruitment, organization and maturation of lymphoid aggregates can provide novel targets for therapeutics and potentially augment vaccines inducing local mucosal immunity.

The defects of EO^-^ mice in cytokine production, lymphocytes recruitment, and lymphocyte organization into aggregates suggest eosinophils may be important in the efficient generation of local lymphoid structures in which lymphocytes are activated and develop. Eosinophils possess Toll-Like Receptors [85,86,87,88,89], indicating that they harbor receptors to respond to infections and trigger the required immune response. This response could be relatively direct; for example, eosinophils might directly present antigens to T cells, as they do during parasitic [24,25,90], viral [48], and allergic reactions [91], indicating they have the required machinery to respond to bacterial infections and activate the subsequent inflammatory response and augment the following adaptive responses.

*Bordetella* spp. provide an exceptional natural-host experimental system in mice, where aspects of bacterial–host interactions can be probed at the molecular level. However, *Bordetella* spp. are not the only pathogens well-adapted to manipulate respiratory immunity to persist. The revelation that eosinophils can mediate effective protective immunity leads to the expectation that successful respiratory pathogens have been under strong selection for the ability to disrupt these functions. It appears likely that one or more human pathogens that were previously common can similarly suppress eosinophil functions, and that their partial or complete control (via medicines, vaccines, and “hygiene”) would likely leave eosinophils tending to be excessively activated, potentially contributing to the upsurge in asthma and allergy. The mechanism by which *B. bronchiseptica* interferes with eosinophil recruitment and function could be an important target for the development of novel therapeutics against eosinophil-mediated pathologies like asthma and allergy. Conversely, understanding how eosinophils contribute to the generation of protective immunity could also facilitate the development of more effective vaccines inducing local mucosal immunity and therapies against a variety of respiratory pathogens.

## 5. Conclusions

It is well-established that *Bordetella* spp. sense details of the local host environment [65,92,93,94,95] and carefully and coordinately regulate multiple mechanisms to modulate host immune responses, suppressing the generation of robust immunity [55,92,96]. The type 3 and type 6 secretion systems [97,98], adenylate cyclase toxin [99,100,101], pertussis toxin [102,103], and other immunomodulators can affect immune cells in vitro and in vivo, and disrupting their carefully choreographed expression can lead to much more robust adaptive immunity [55]. Our previous work revealed how classical *Bordetella* spp. (*B. pertussis*, *B. parapertussis*, and *B. bronchiseptica*) similarly respond to blood and serum [65], and one of the genes that increase expression levels in response to blood and serum is *btrS* (*sigE* type sigma factor). When *btrS* is knocked out, *B. bronchiseptica* fails to dampen the host response and induces much stronger protective immunity [55]. Here we reveal that *btrS* is involved in the blockage of eosinophil recruitment to the lungs, reducing adaptive immune responses and promoting persistence. Moreover, our data indicate that eosinophils mediate cytokine production and B-cell responses to respiratory infections, providing new roles for these cells in the generation of protective respiratory immunity. Excitingly, our work associates eosinophils with the generation of lymphoid aggregates, suggesting that a better understanding of the underlying mechanisms can provide targets for therapeutics against a broad spectrum of diseases. There is also a positive correlation between *Bordetella* spp. infections and the development of asthma and allergy [104,105,106,107,108,109,110,111], highlighting the clinical significance of these findings and suggesting there may be more implications in this complex interaction.

## Figures and Tables

**Figure 1 microorganisms-08-01808-f001:**
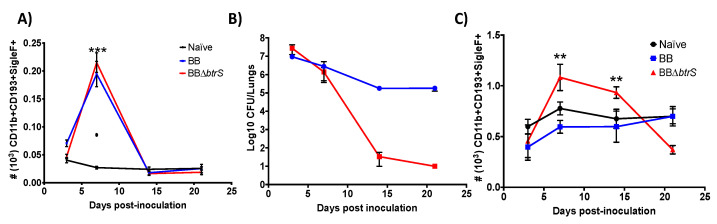
Eosinophils numbers increase in the spleen, but *btrS* blocks their recruitment to lungs. C57BL/6J mice were intranasally inoculated with PBS (black), or PBS containing 5 × 10^5^ CFU of *Bb* (blue) or *Bb*Δ*btrS* (red). Mice were euthanized at different times, to enumerate CFU (**B**) and eosinophils CD11b^+^CD193^+^SiglecF^+^ in the spleen (**A**) and the lungs, CD11b^+^CD193^+^SiglecF^+^ (**C**). The graphs show the average and SEM. ** *p* < 0.001 and *** *p* < 0.0001, using two-way ANOVA test *N* = 10.

**Figure 2 microorganisms-08-01808-f002:**
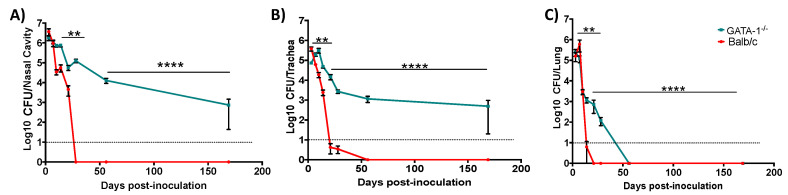
Eosinophil-deficient mice failed to clear bacterial infection from the respiratory tract. Balb/cJ—WT (wild-type) (red) or DdblGATA-1—EO^-^ (eosinophil-deficient) (green) mice were intranasally challenged with 50 μL of PBS containing 5 × 10^5^
*Bb*D*btrS*. At different days, mice were euthanized, and colonies were enumerated in the nasal cavity (**A**), trachea (**B**), or lungs (**C**). The graphs show the average and SEM. ** *p* < 0.01, **** *p* < 0.0001, using two-way ANOVA test. This experiment was conducted multiple independent times, to compensate for possible effects of times of the year and spontaneous allergic reactions on eosinophils counts. *n* = 22.

**Figure 3 microorganisms-08-01808-f003:**
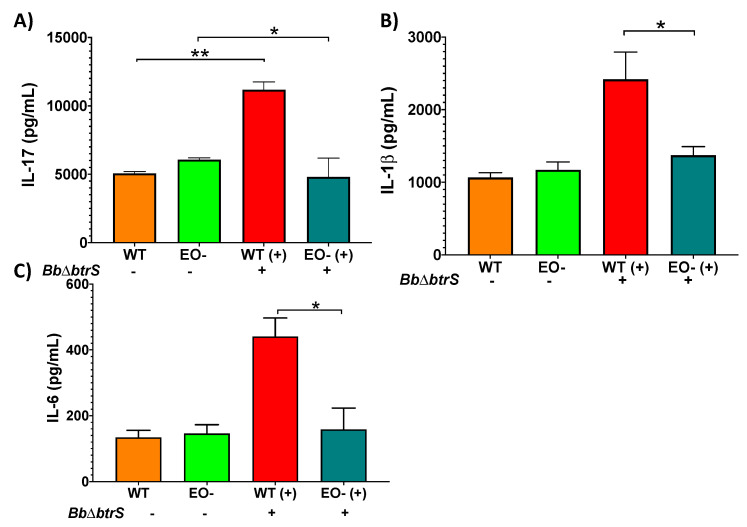
Eosinophil-deficient mice failed to secrete IL-17, IL-1β, and IL-6. Groups of mice were intranasally challenged with 50 μL of PBS, or PBS containing 5 × 10^5^
*Bb*D*btrS*; they were euthanized on day 7 (**A**,**B**) or 14 (**C**) post-challenge, and trachea was collected and homogenized, to perform ELISA. The graphs show the average and SEM. * *p* < 0.05 and ** *p* < 0.01, using two-way ANOVA. *n* = 6.

**Figure 4 microorganisms-08-01808-f004:**
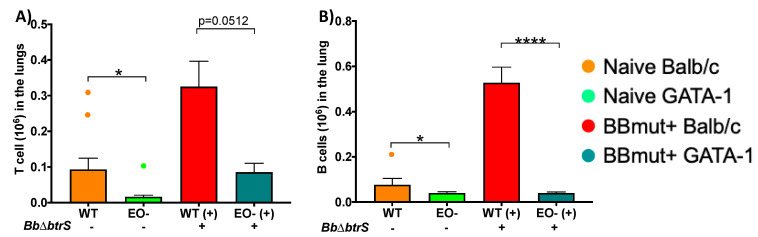
Eosinophil-deficient mice failed to recruit lymphocytes to the lung. WT or EO^-^ mice were intranasally challenged with 50 μL of PBS, or PBS containing 5 × 10^5^
*Bb*D*btrS*. Mice were euthanized at day 14 post-infection, to enumerate T (**A**) and B (**B**) cell populations. WT indicates Balb/c, and EO- stands for DdblGATA-1 mice. BbD*btrS* indicates if intranasal challenge has been (+) or not (-) performed. The graphs show the average and SEM. * *p* < 0.05, **** *p* < 0.001, using two-way ANOVA. This experiment was conducted multiple independent times, to control for possible effects of times of the year and spontaneous allergic reactions. *n* = 20.

**Figure 5 microorganisms-08-01808-f005:**
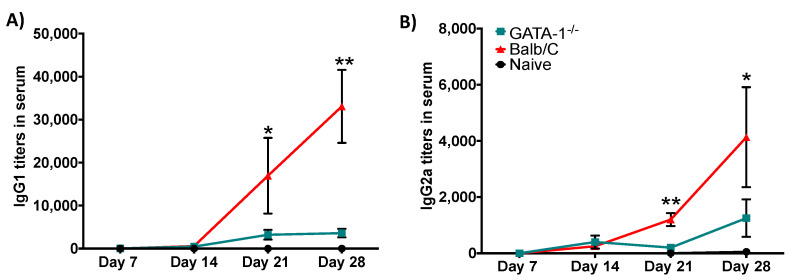
Eosinophil-deficient mice failed to generate protective antibodies. WT or EO mice were intranasally challenged with 50 μL of PBS or PBS containing 5 × 10^5^
*Bb*D*btrS*, in two independent experiments. At different days, mice were euthanized, and serum was collected to perform ELISA, to evaluate anti-Bb antibody titers (results are the average of two independent assays in triplicate). IgG1 (**A**) and IgG2a (**B**) were evaluated. The data are shown as average and SEM; * *p* < 0.05 and ** *p* < 0.01, using Student *t*-test. *n* = 6.

**Figure 6 microorganisms-08-01808-f006:**
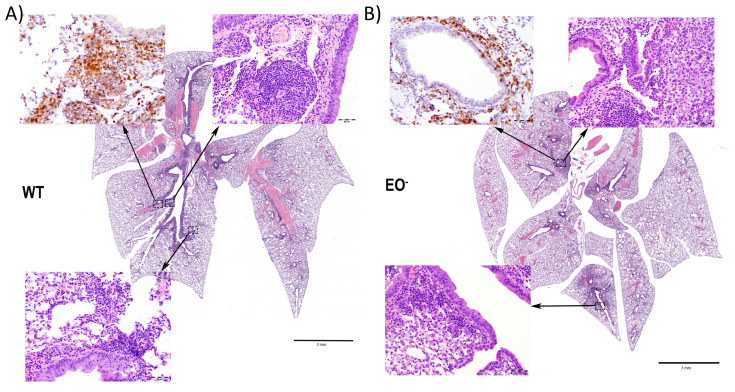
Eosinophil-deficient mice were deficient in generating organized BALT. Groups of 6 Balb/cJ (WT) (**A**) and GATA-1 (EO^-^) (**B**) mice were intranasally challenged with 50 μL of PBS containing 5 × 10^5^
*Bb*D*btrS*, and, at day seven post-challenge, lungs were fixed in 10% formalin solution for hematoxylin and eosin staining. Scale bar = 3 mm. *n* = 6.

**Table 1 microorganisms-08-01808-t001:** List of antibodies for flow cytometry.

Flow Cytometry Antibodies				
Fluorochrome	Target	Clone	Vendor	Ref
APC-Cy7	GR1	RB6–8C5	Biolegend	108–424
PE	CD11b	M1/70	Tonbo	50–0112–U100
PerCP	F4/80	BM8	Biolegend	123,126
APC	SiglecF	S17007L	Biolegend	155,508
BV510	CD193	CCR3	BD Biosciences	747,820
PerCP	CD19	6D5	Biolegend	115,532
PE	CD90.2	53–2.1	Biolegend	140,308
AF488	CD4	GK1.5	Biolegend	100,423
BV510	CD8	53–6.7	BD Biosciences	563,068

**Table 2 microorganisms-08-01808-t002:** Gating strategy.

	+	+	+
Eosinophils	CD11b	CD193	SiglecF
B cells	CD11b	CD19	
CD4	CD11b	CD90.2	CD4
CD8	CD11b	CD90.2	CD8

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
