# Peer review of "Disrupting Bordetella Immunosuppression Reveals a Role for Eosinophils in Coordinating the Adaptive Immune Response in the Respiratory Tract"

_microorganisms, 2020, doi:10.3390/microorganisms8111808_

Round 1

Reviewer 1 Report

I thank the authors for the improvements they have made to the manuscript. 

There are still some issues to be resolved:

  • Figure 1 is missing, so, it is not possible to provide comments regarding this figure
  • In figure 4, please provide statistical analysis comparing the naive WT (orange) and naive EO- (green) T cell and B cell numbers. If there are significantly lower numbers of these cells in the resting lungs of EO- animals, some discussion of how this may occur and its impact on infection should be included 
  • Line 178 refers to figure 1, should this reference Fig 4 instead?
  • There are no details provided for the IHC included in figure 6, please include a description in the figure legend, a reference to the outcome in the results section and a protocol in the methods section
  • Line 215, there is specific mention of macrophages, neutrophils and lymphocyte accumulations, provide some information on how these cells were identified or quantified

Author Response

I thank the authors for the improvements they have made to the manuscript.

We would like to thank the reviewer for the previous suggestions that have significantly improved this manuscript. We are thankful for the extra-comments here provided that will continue to develop the manuscript.

There are still some issues to be resolved:

Figure 1 is missing, so, it is not possible to provide comments regarding this figure

We apology for the inconvenience, we have now confirmed that Fig 1 is included in the manuscript uploaded on the submission system.

In figure 4, please provide statistical analysis comparing the naive WT (orange) and naive EO- (green) T cell and B cell numbers. If there are significantly lower numbers of these cells in the resting lungs of EO- animals, some discussion of how this may occur and its impact on infection should be included

This is a great suggestion and we would like to thank the reviewer for this comment. The role of eosinophils as modulators of B and T cells has been reported (“Eosinophils Regulate Peripheral B Cell Numbers in Both Mice and Humans” and “Immunoregulatory roles of eosinophils: a new look at a familiar cell”) but our results provide further insight into the effects of eosinophils during adaptive responses to bacterial infections. We have now included the statistics on B and T cells comparing uninfected Balb/c and GATA-1, and we discussed in the context of comparing our data with previously published work.

Line 178 refers to figure 1, should this reference Fig 4 instead?

Line 178 is correctly referring to Figure 1.

There are no details provided for the IHC included in figure 6, please include a description in the figure legend, a reference to the outcome in the results section and a protocol in the methods section

We have added the details of the procedure in Material and Methods.

Line 215, there is specific mention of macrophages, neutrophils, and lymphocyte accumulations, provide some information on how these cells were identified or quantified

These cells were identified by our certified pathologist using IHC and this is now described in the text.

Reviewer 2 Report

Review of Microorganisms 968406

The authors have submitted a revision to the manuscript now titled “Disrupting Bordetella immunosuppression reveals a role for eosinophils in coordinating the adaptive immune response in the respiratory tract” wherein they suggest that btrS is used by bacteria to modulate eosinophil responses in the host. Although the authors have updated references, and changed some of the Figures, the manuscript still has significant weaknesses in scientific rigor, premise, and the conclusions that the authors have drawn are not accurate based on the data provided. Additionally, failure to show the wild-type bacterial strain side-by-side for comparison suggests that the authors did not use the wild-type strain for studies in tandem – this is a significant and fundamental weakness when using a live pathogen.

1) The authors have now included more references from the eosinophil field – however, the inclusion of these papers in relation to the authors’ statements are not always correct and therefore, a weakness. Furthermore, language editing is necessary throughout the manuscript. Authors refer to eosinophil ‘hyperreactivity’ and ‘hyperactivity’ – it is unclear what is meant as eosinophil functions cannot be described as either.

2) Figure 1: (a) The authors have now changed the scales in the Figure for spleen and lung eosinophils noted as CD11b+CCR3+Sig-F+ cells. Now, there are more eosinophils in the spleen than the lungs – not possible as the lungs are the infection site. In fact, the lungs are said to have as little as 100 eosinophils – this is also experimentally difficult to ascertain by flow and unreliable at such small numbers especially with these markers which are altered on the eosinophil surface based on environment. Gating strategy not provided nor are flow cytometry controls to determine data accuracy. Based on the table provided, there is significant overlap between the three fluorochromes selected for eosinophil markers which would make it impossible to separate these cells! This as an example that flaws are likely in the experimental setup. (b) Bacterial titer and eosinophil infiltration pattern into the lungs do not correlate. A reduction in titer occurs a week after the said peak in eosinophils at which point eosinophils are unlikely to contribute as they would be dead. Also, how <400 eosinophils will reduce such a high pathogen dose is also not clear. (c) If the wild-type Bb (blue line) inhibits eosinophils, then there should be clear differences in eosinophil numbers between the red (mutant Bb) and blue lines in both niches – this is also not observed. In fact, eosinophil numbers should always remain higher in the mutant Bb infection, which is not the case. Also, why are eosinophil numbers increasing in naïve mice? This should not happen at steady-state. *** Cumulatively, the authors conclusion that eosinophil ‘recruitment’ is hindered by btrS is not supported.

3) Figure 2: While these data do support a role for eosinophil-mediated antibacterial functions, the authors do not provide data with the wild-type Bb and point the reader to a previous publication instead. This suggests that the wild-type Bb was not run in parallel with the mutant strain in the GATA null mice. This is a significant weakness in the scientific rigor for several reasons: (a) When using any live infectious agent, all controls should be run in tandem in every experiment as it is a dynamic biological system with the pathogen and host. (b) There can be batch variations in pathogen virulence. (c) B6 mice (Fig 1) and BALB/c mice (remaining Figures) are significantly different in immune bias – yet another reason to include wild-type Bb in the experiments. (d) Additionally, eosinophil numbers in WT BALB/c mice should also be provided to validate hypothesis that eosinophil recruitment is higher in mutant-Bb infections. *** Therefore, definite conclusions cannot be drawn regarding the necessity of eosinophils to clear bacteria.

4) Figure 3: My previous comment has not been fully addressed here. The authors draw conclusions on adaptive immune responses based on three cytokines that are measured at early timepoints in the trachea. Data should be represented as the mean and SD instead of SEM, and it is unclear how 2-way ANOVA was possible as analysis should be by 1-way ANOVA with multiple comparisons test. Why were difference timepoints selected for cytokine measurement? Again, the bacterial titers and cytokines levels do not correspond. As stated before, the experimental rigor is weak here in the absence of the wild-type Bb strain use as controls.  Again, as mentioned before the authors’ conclusion that eosinophils regulate cytokine production is somewhat premature as they have not shown what other immune cells are impacted in the GATA null mice during this bacterial infection.

5) Figure 4: Based on the table provided overlapping fluorochromes have been selected for CD4 and CD8 and controls used for flow experiments not provided. Based on these fluorochromes, it would be impossible to differentiate between T cell types. In their response, the authors refer to a previous publication in which different fluorochromes were used for T cells. In any case, each study should be a stand-alone document with rigorously performed experiments. One cannot refer to previous controls published before in biological systems. Naïve, uninfected animals have ~200,000 lymphocytes – like for eosinophils as mentioned above, this is not possible at steadystate.

6) Figure 5: Similar to my comment above, the authors have not addressed my point. Further, total Ig levels are provided which show an increase at late timepoints. Again, SEM is used incorrectly instead of SD. Serum Ig levels are measured for a lung infection. The authors argue that tracheal cytokines are more relevant, and yet, measure serum Igs and show lung histology. Varying timepoints are selected for different analyses.

7) Figure 6: My previous comments have not been fully addressed. There is still no proof that these aggregates are BALTs. The IHC shows different areas in each sample – so not adequate for comparison. Also why not show the whole section for the reader to note if there may be a reduction in CD3+ cells? Why not do quantification if there were 6 animals in each group?

8) AS previously stated, significant weaknesses in the data preclude the formulation of conclusions. Sweeping conclusions are made without direct evidence. (a) adaptive immunity not measured directly and robustly. (b) BALTs not proven. In the Discussion the authors mention that BALTs were noted ‘near the infection site’ and yet, state in their response that trachea are more relevant. (c) Authors’ naivete is evident with incorrect citations in the Introduction and their conclusions that antibacterial functions and immunomodulatory functions are ‘new’ discoveries for eosinophils – they have been well known for these functions for some time. "Anamnestic" responses are referred to in several places when the study is a single agent model. (d) TLRs are used to describe antigen presentation! (e) Discussion of hygiene hypothesis is ambiguous here as eosinophils have no memory response.

Author Response

The authors have submitted a revision to the manuscript now titled “Disrupting Bordetella immunosuppression reveals a role for eosinophils in coordinating the adaptive immune response in the respiratory tract” wherein they suggest that btrS is used by bacteria to modulate eosinophil responses in the host. Although the authors have updated references, and changed some of the Figures, the manuscript still has significant weaknesses in scientific rigor, premise, and the conclusions that the authors have drawn are not accurate based on the data provided. Additionally, failure to show the wild-type bacterial strain side-by-side for comparison suggests that the authors did not use the wild-type strain for studies in tandem – this is a significant and fundamental weakness when using a live pathogen.

We are sorry to hear that the reviewer is not enthusiastic about our findings and we hope this revised version is more inspiring. We have to emphasized that we decided to not include the wildtype RB50 in our studies because no differences in colonization were identified between the mutant DdblGATA-1 mice and wild type Balb/c, and assays in these mice have been included in many previous papers. Showing these results again is neither new nor important to the central goal of this manuscript. Our goal is to build on our recently published evidence that B. bronchiseptica disrupts the immune response via BtrS-mediated modulation, and to reveal the remarkable effects that eosinophils can have when not suppressed.  We are thankful that all three reviewers appreciate that and believe the focus on the comparison to wild type is based on rigorous training that simply does not apply in this instance.

1) The authors have now included more references from the eosinophil field – however, the inclusion of these papers in relation to the authors’ statements are not always correct and therefore, a weakness. Furthermore, language editing is necessary throughout the manuscript. Authors refer to eosinophil ‘hyperreactivity’ and ‘hyperactivity’ – it is unclear what is meant as eosinophil functions cannot be described as either.

We appreciate the opportunity to improve the clarity and have now revised the references and the English has been carefully edited. As the reviewer has suggested, Hyperreactivity and Hyperactivity have been replaced by dysregulation and hyperresponsiveness following the nomenclature of previous publications (Gundel, R. H.; Letts, L. G.; Gleich, G. J., Human eosinophil major basic protein induces airway constriction and airway hyperresponsiveness in primates. J Clin Invest 1991, 87 (4), 1470-3.)

2) Figure 1: (a) The authors have now changed the scales in the Figure for spleen and lung eosinophils noted as CD11b+CCR3+Sig-F+ cells. Now, there are more eosinophils in the spleen than the lungs – not possible as the lungs are the infection site. In fact, the lungs are said to have as little as 100 eosinophils – this is also experimentally difficult to ascertain by flow and unreliable at such small numbers especially with these markers which are altered on the eosinophil surface based on environment. Gating strategy not provided nor are flow cytometry controls to determine data accuracy. Based on the table provided, there is significant overlap between the three fluorochromes selected for eosinophil markers which would make it impossible to separate these cells! This as an example that flaws are likely in the experimental setup. (b) Bacterial titer and eosinophil infiltration pattern into the lungs do not correlate. A reduction in titer occurs a week after the said peak in eosinophils at which point eosinophils are unlikely to contribute as they would be dead. Also, how <400 eosinophils will reduce such a high pathogen dose is also not clear. (c) If the wild-type Bb (blue line) inhibits eosinophils, then there should be clear differences in eosinophil numbers between the red (mutant Bb) and blue lines in both niches – this is also not observed. In fact, eosinophil numbers should always remain higher in the mutant Bb infection, which is not the case. Also, why are eosinophil numbers increasing in naïve mice? This should not happen at steady-state. *** Cumulatively, the authors conclusion that eosinophil ‘recruitment’ is hindered by btrS is not supported.

We would like to thank the reviewer for this comment. We have provided our data set to an external expert that has reanalyzed the data in a blind fashion manner. He has suggested to draw separately the outlawyers that were affecting our data presentation. We are very thankful for this comment and the opportunity to improve the data presentation.

3) Figure 2: While these data do support a role for eosinophil-mediated antibacterial functions, the authors do not provide data with the wild-type Bb and point the reader to a previous publication instead. This suggests that the wild-type Bb was not run in parallel with the mutant strain in the GATA null mice. This is a significant weakness in the scientific rigor for several reasons: (a) When using any live infectious agent, all controls should be run in tandem in every experiment as it is a dynamic biological system with the pathogen and host. (b) There can be batch variations in pathogen virulence. (c) B6 mice (Fig 1) and BALB/c mice (remaining Figures) are significantly different in immune bias – yet another reason to include wild-type Bb in the experiments. (d) Additionally, eosinophil numbers in WT BALB/c mice should also be provided to validate hypothesis that eosinophil recruitment is higher in mutant-Bb infections. *** Therefore, definite conclusions cannot be drawn regarding the necessity of eosinophils to clear bacteria.

The first experiments of the manuscript (figure 1 and figure S1) included wild type (RB50) in  parallel with our mutant strain. These experiments did not identify any defect in the mice challenged with the wildtype (Fig S1), but identify a defect for our mutant bacteria that disappears in GATA null mice. Using this mutant bacteria, we then probe the specific defect in GATA null mice.  The lack of phenotype for the wild type bacteria is not the focus, would not reveal anything novel and is a waste of animals, time and effort.

4) Figure 3: My previous comment has not been fully addressed here. The authors draw conclusions on adaptive immune responses based on three cytokines that are measured at early timepoints in the trachea. Data should be represented as the mean and SD instead of SEM, and it is unclear how 2-way ANOVA was possible as analysis should be by 1-way ANOVA with multiple comparisons test. Why were difference timepoints selected for cytokine measurement? Again, the bacterial titers and cytokines levels do not correspond. As stated before, the experimental rigor is weak here in the absence of the wild-type Bb strain use as controls.  Again, as mentioned before the authors’ conclusion that eosinophils regulate cytokine production is somewhat premature as they have not shown what other immune cells are impacted in the GATA null mice during this bacterial infection.

Regarding data presentation, we consider SEM to be more statistically accurate as the SEM quantifies how precisely you know the true mean of the population. It takes into account both the value of the SD and the sample size. Contrary, standard deviation (SD) measures the amount of variability, or dispersion, from the individual data values to the mean, while the standard error of the mean (SEM) measures how far the sample mean of the data is likely to be from the true population mean.

In regard to the question of why we performed the measurement of cytokines from trachea. We consider that this is an accurate measurement of the inflammatory response to Bordetella infections. Although we understand that we could have measured all in one place, we decided to use all the lung to evaluate immune cell populations and use the trachea of the same mice to enumerate cytokines. Moreover, we also used the blood of the same mice to enumerate antibody titers. We use the same individuals for all measurements, and we believe that this is correct.

5) Figure 4: Based on the table provided overlapping fluorochromes have been selected for CD4 and CD8 and controls used for flow experiments not provided. Based on these fluorochromes, it would be impossible to differentiate between T cell types. In their response, the authors refer to a previous publication in which different fluorochromes were used for T cells. In any case, each study should be a stand-alone document with rigorously performed experiments. One cannot refer to previous controls published before in biological systems. Naïve, uninfected animals have ~200,000 lymphocytes – like for eosinophils as mentioned above, this is not possible at steadystate.

We would like again to thank the reviewer for this comment. We have provided our data to an external Flow technician, who has performed a new analysis in a blind fashion manner and has suggested to draw separately the outlawyers that were affecting our data presentation. Please see new figure 4.

6) Figure 5: Similar to my comment above, the authors have not addressed my point. Further, total Ig levels are provided which show an increase at late timepoints. Again, SEM is used incorrectly instead of SD. Serum Ig levels are measured for a lung infection. The authors argue that tracheal cytokines are more relevant, and yet, measure serum Igs and show lung histology. Varying timepoints are selected for different analyses.

Regarding data presentation, please see the response above (#4) and just briefly, our decision was based on the fact that standard deviation (SD) measures the amount of variability, or dispersion, from the individual data values to the mean, while the standard error of the mean (SEM) measures how far the sample mean of the data is likely to be from the true population mean.

In regard to the measurement of the antibody titers on serum, this measurement is more relevant as it is the standard to measure antibody titers on serum, making our results comparable to another Bordetella spp. literature.

7) Figure 6: My previous comments have not been fully addressed. There is still no proof that these aggregates are BALTs. The IHC shows different areas in each sample – so not adequate for comparison. Also why not show the whole section for the reader to note if there may be a reduction in CD3+ cells? Why not do quantification if there were 6 animals in each group?

We have changed the statement about BALT due to the reviewer concerns. The pathology was performed in groups of mice and the photo is representative of the general findings.

The pathologist, who is a certified vet pathologist, recommended to not use numerical scoring and he explained “The lung is a 3D structure, think about it as the measurements of an orange. In the top it can be very small compared with a measure of the same orange in the middle”.

For all the pathology, we entrusted a professional with our blinded samples for the analysis as well as we have asked him to do the more relevant and pertinent studies to answer our questions.

8) AS previously stated, significant weaknesses in the data preclude the formulation of conclusions. Sweeping conclusions are made without direct evidence. (a) adaptive immunity not measured directly and robustly. (b) BALTs not proven. In the Discussion the authors mention that BALTs were noted ‘near the infection site’ and yet, state in their response that trachea are more relevant. (c) Authors’ naivete is evident with incorrect citations in the Introduction and their conclusions that antibacterial functions and immunomodulatory functions are ‘new’ discoveries for eosinophils – they have been well known for these functions for some time. "Anamnestic" responses are referred to in several places when the study is a single agent model. (d) TLRs are used to describe antigen presentation! (e) Discussion of hygiene hypothesis is ambiguous here as eosinophils have no memory response.

We have changed the statements aforementioned (c,d,e). We hope that the reviewer finds this version of the manuscript more appealing. We have further clarified that the focus of this manuscript is that, disrupting the bacteria’s ability to manipulate host immune responses has allowed us to identify this important role for eosinophils during bacterial respiratory infections. This finding emphasizes that we might not correlate eosinophils with bacterial infections because eosinophils may be being blocked by pathogens. These findings are likely to be relevant to many other pathogens, especially within the respiratory tract.

Reviewer 3 Report

Bordetella bronchiseptica is an important respiratory pathogen and model of B. pertussis. Understanding the immune response to B. bronchiseptica and by association to B. pertussis is critical if Whooping cough is going to be successfully eliminated from the population. The manuscript proposes a novel role for eosinophils in the protection against respiratory methods and shows how B. bronchiseptica subverts this line of defence. The hypothesis of this manuscript is sound, and the appropriate experiments were used to test it. Overall, this is a well-designed study that sheds new light onto the immune response to Bordetella spp. and could potentially shift our understanding of Bordetella infection.

Major:

Figure 1 was missing from my copy, so I’m unable to comment on it.

Minor:

Figure 3 and 4, Greek characters are missing in the axes labels. The X axis labels aren’t clear.

Line 193: Should read ‘Student’s t-test’ not ‘T-Test’.

Author Response

Bordetella bronchiseptica is an important respiratory pathogen and model of B. pertussis. Understanding the immune response to B. bronchiseptica and by the association to B. pertussis is critical if Whooping cough is going to be successfully eliminated from the population. The manuscript proposes a novel role for eosinophils in the protection against respiratory methods and shows how B. bronchiseptica subverts this line of defense. The hypothesis of this manuscript is sound, and the appropriate experiments were used to test it. Overall, this is a well-designed study that sheds new light onto the immune response to Bordetella spp. and could potentially shift our understanding of Bordetella infection.

 We would like to thank the reviewer for the very nice words and the passion for our work. We hope that this revised and improved version is exciting for the reviewer.

Major:

Figure 1 was missing from my copy, so I’m unable to comment on it.

 Fig 1 is included in the updated version. We apology for the inconvenience.

Minor:

Figures 3 and 4, Greek characters are missing in the axes labels. The X-axis labels aren’t clear.

We apology for the inconvenience and we have now uploaded new figures with revised legends to improve clarity.

Line 193: Should read ‘Student’s t-test’ not ‘T-Test’.

We have change t-test as requested in figure 5 as requested.

This manuscript is a resubmission of an earlier submission. The following is a list of the peer review reports and author responses from that submission.

Round 1

Reviewer 1 Report

In this manuscript titled “Bacterial blockage of eosinophil recruitment to the lungs mediate long-term persistence”, the authors discuss the possibility that invading bacteria block eosinophil recruitment into the lungs thereby limiting the magnitude of the antibacterial host immune response. While this is an interesting concept for which the data provide some credence to, weaknesses in the experimental design, significant problems with the presented data, and limitations in the scientific premise all significantly lessen enthusiasm for this manuscript.

1) It is apparent that the authors have limited experience working with eosinophils. The introduction has failed to cite a number of papers in the field of respiratory infectious diseases that have demonstrated a role for antiviral and antibacterial functions for eosinophils. The scientific premise for the focus on eosinophils in this model is weak.

2) Figure 1: Fig 1B shows that the mutant bacterial strain does not replicate as efficiently in the host as the WT strain. These data are support a role for BtrS as a virulence factor. However, in Figure 1A, the authors show that the number of eosinophils in the spleen peaks at day 7 before flattening out. These data are concerning for a number of reasons: There are about 50 million eosinophils in the spleens according to this graph in Fig 1A even in naïve mice. This is not the case – mice have about a total of 1×108 cells in the spleen, of which, only about 107 can be harvested, of which, <1% are eosinophils at steady state. According to Fig 1A, after infection, the number of eosinophils increase to 2×108 which is impossible. Similar problems exist in Fig 1C in which the authors states that over 1×109 eosinophils are recruited in a bacterial infection – this is also impossible for similar reasons as above. Therefore, the data, as presented suggest a significant flaw in the methods employed. It is also unclear how the authors arrived at these numbers as they state that only 2 million cells were stained.

3) Figure 2: Here the data support a role for the eosinophils in bacterial clearance. The data with the WT bacterial strain has not been provided here, but important for comparison.

4) Figure 3: The authors state their interest in adaptive immune responses, but have measured cytokines that are important in the acute phase of infection. While it is clear that the eosinophil deficient animals had reduced cytokines, the authors conclusion that eosinophils regulate cytokine production is somewhat premature as they have not shown what other immune cells are impacted in the GATA null mice during this bacterial infection. Additionally, the authors have not shown the use of these mice and their WT counterparts with the wt strain of bacteria for comparison. Why are cytokines measured in the trachea and not in the lung?

5) Figure 4: Similar issue exists with the void in data that show the results with the WT bacterial strain and the number of T and B cells are rather high as shown in the Figure for naïve animals with ~200,000 T cells etc. As is the case with eosinophils, the authors have not provided a staining method for flow, but simply cite other papers. The controls used and the gating strategies and antibody cocktails used are not provided.

6) Figure 5: The WT bacterial strain should be shown for comparison.

7) Figure 6: The histology for naïve animals and those that were infected with the WT strain should be provided for comparison. There is no major difference in the lung inflammation between the two strains of mice based on Fig 6. In the absence of immunohistochemistry as proof, it is difficult to state with certainty that the foci are BALTs.

8) Significant weaknesses in the data preclude the formulation of conclusions.

Reviewer 2 Report

The manuscript “Bacterial blockage of eosinophil recruitment to the lungs mediate long-term persistence” is well written and describes an interesting virulence mechanism utilized by Bordetella bronchiseptica. This study uses robust models to demonstrate btrS-mediated impairment of eosinophil-promoted bacterial clearance and speculates that eosinophils are a central regulator of immune responses to respiratory pathogens. The work will contribute significantly to the field. Suggested improvements are outlined below.

Major comments:

  1. Section 2.5, It is put forward that eosinophils mediate the recruitment of lymphocytes to the infected lungs as demonstrated by lower numbers of T cells and B cells in the lungs of infected GATA-1-/- mice compared with infected Balb/c mice. However, from the data provided, it is not possible to determine whether the cell deficit is the result of reduced numbers systemically or reduced recruitment to the lungs. E.g. eosinophil deficiencies have previously been associated with reduced numbers of plasma cells in the bone marrow and gut (Chu, et al., Eosinophils are required for the maintenance of plasma cells in the bone marrow. Nat. Immunol. 2011. 12: 151–159. Chu, et al., Eosinophils promote generation and maintenance of immunoglobulinAexpressing plasma cells and contribute to gut immune homeostasis. Immunity 2014). Please rephrase the description of this result or provide data demonstrating no impairment in systemic or lymphoid tissue-associated lymphocyte numbers.
  2. Section 2.7, please provide a quantitative or semi-quantitative analysis of the number or size of BALT induced in Balb/c vs GATA-1-/- mice to support the single sample image provided. In addition, ideally the BALT described in BBbtrS-infected Balb/c mice would be verified by IHC of IF to demonstrate the presence of distinctly arranged T and B cells in the inflamed areas. Otherwise, the pathology is best described as lymphoid aggregates.
  3. All experiments performed in the eosinophil-deficient mice feature infection with the BBbtrS strain. Given that btrS-expressing RB50 fails to recruit eosinophils to the lungs, an experiment demonstrating no change in RB50 bacterial burden or pathogenesis would greatly strengthen the hypothesis that these eosinophils directly contribute to BBbtrS clearance.
  4. Earlier work by this group demonstrated that mice infected with BBbtrS also had increased recruitment of neutrophils, macrophages, T cells and B cells compared with RB50-infected mice. For completeness, this previous observation should be discussed.

Minor comments:

  1. The introduction could be improved with more discussion of the known roles of eosinophils in bacterial infection.
  2. Provide a reference for the work discussed in paragraph 3 of the introduction.
  3. The arrangement and order of discussion of Fig. 1 is somewhat confusing and may be improved by swapping Fig 1A and Fig 1B.
  4. Line 139, superscript required for “5x105 CFU”
  5. Line 153, “eosinophil numbers increased in both the spleen and lungs”. There is significance indicated for the numbers of eosinophils in the spleen (Fig 1A) at peak infection, but no significance is indicated on the lung analyses (Fig 1C). Please provide information on the statistical significance of the lung eosinophil numbers at peak infection for each group.
  6. Line 176, sentence is incomplete.
  7. Fig 3 & 4, please change the key nomenclature to BBbtrS to keep consistent with the manuscript.
  8. Fig 5, indicate the infected groups in the key or describe the groups, naïve etc. in the legend.
  9. Line 374, provide the full name for AVDL
  10. Section 5.3, please provide details on how the samples were acquired for assessment of antibody titer.